**Data Availability Statement:** All relevant data and reference to them are within the paper and its Supporting Information files.

# The leucine-NH4+ uptake regulator Any1 limits growth as part of a general amino acid control response to loss of La protein by fission yeast

Vera Cherkasova[1], James R. Iben[2], Kevin J. Pridham[3], Alan C. Kessler[4], Richard J. Maraia[4]*

1 Kelly@DeWitt, Inc, National Library of Medicine, National Institutes of Health, Bethesda, MD, United States of America, 2 Molecular Genomics Core, Division of Intramural Research, Eunice Kennedy Shriver National Institute of Child Health and Human Development, National Institutes of Health, Bethesda, MD, United States of America, 3 Fralin Biomedical Research Institute at Virginia Tech, Roanoke, VA, United States of America, 4 Section on Molecular and Cell Biology, Division of Intramural Research, Eunice Kennedy Shriver National Institute of Child Health and Human Development, National Institutes of Health, Bethesda, MD United States of America

* maraiar@nih.gov

## Abstract

The *sla1+* gene of *Schizosachharoymces pombe* encodes La protein which promotes proper processing of precursor-tRNAs. Deletion of *sla1* (*sla1Δ*) leads to disrupted tRNA processing and sensitivity to target of rapamycin (TOR) inhibition. Consistent with this, media containing NH4+ inhibits leucine uptake and growth of *sla1Δ* cells. Here, transcriptome analysis reveals that genes upregulated in *sla1Δ* cells exhibit highly significant overalp with general amino acid control (GAAC) genes in relevant transcriptomes from other studies. Growth in NH4+ media leads to additional induced genes that are part of a core environmental stress response (CESR). The *sla1Δ* GAAC response adds to evidence linking tRNA homeostasis and broad signaling in *S. pombe*. We provide evidence that deletion of the Rrp6 subunit of the nuclear exosome selectively dampens a subset of GAAC genes in *sla1Δ* cells suggesting that nuclear surveillance-mediated signaling occurs in *S. pombe*. To study the NH4+-effects, we isolated *sla1Δ* spontaneous revertants (SSR) of the slow growth phenotype and found that GAAC gene expression and rapamycin hypersensitivity were also reversed. Genome sequencing identified a F32V substitution in Any1, a known negative regulator of NH4+-sensitive leucine uptake linked to TOR. We show that 3H-leucine uptake by SSR-*any1-F32V* cells in NH4+-media is more robust than by *sla1Δ* cells. Moreover, F32V may alter *any1+* function in *sla1Δ* vs. *sla1+* cells in a distinctive way. Thus deletion of La, a tRNA processing factor leads to a GAAC response involving reprogramming of amino acid metabolism, and isolation of the *any1-F32V* rescuing mutant provides an additional specific link.

**Funding:** I would like to note that this work was supported by the Division of Intramural Research (HD000412-31 PGD) of the Eunice Kennedy Shriver National Institute of Child Health and Human Development, National Institutes of Health. The funders had no role in study design, data collection, interpretation, or decision to submit for publication.

**Competing interests:** The authors have declared that no competing interests exist.

## Introduction

La is an abundant nuclear protein that binds the UUU-3'OH end motif produced by transcription termination by RNA polymerase III (Pol III) [reviewed in 1]. This protects the nascent transcripts from 3' exonucleases while La functions as a molecular charperone during the multistep process of tRNA maturation prior to nuclear export [1–3]. La is essential in a range of species, but not budding and fission yeast even though aberrant nuclear pre-tRNAs accumulate in its absence [1, 3]. Yeast do not survive without La if an essential tRNA suffers a mutation that impairs the folding, processing or critical modification of its precursor [3–5].

tRNAs are expressed at 15 to 20-fold molar excess relative to ribosomes [6]. Nonetheless increases in Pol III activity result in accumulation of unprocessed and hypomodified tRNA with evidence that their processing, modification and export activities are limiting in yeast [7–9]. Environmental stressors also lead to accumulation of pre-tRNAs [9]. However even in unstressed wild type yeast, large amounts of precursor-tRNAs succumb to degradation by the nuclear exosome, believed to reflect elimination of termination flawed and/or mis-processed species [10, 11]. Aberrant pre-tRNAs are degraded by one of the multiple quality control pathways that target defective RNAs and serve to moderate gene expression and prevent disease [12]. The nuclear surveillance system that targets aberrant pre-tRNAs, begins with 3' oligoadenylation by the Trf4 subunit of the 'TRAMP' complex followed by 3'-exonucleolytic digestion by the Rrp6-nuclear exosome [13–16]. As the nucleolytic attack is directed at the 3'-end, La protein competes with and protects pre-tRNAs, including modestly aberrant ones against such degradation [17–20].

Nuclear surveillance was found associated with a stress response in *S. cerevisiae* that activates the general amino acid control (GAAC) pathway (AAR, amino acid response in mammals) [17, 18, 21–23]. This is distinct from the classic GAAC activation pathway as it does not require the Gcn2 protein kinase. In the classic pathway, Gcn2 senses amino acid deficiency by recognizing uncharged tRNA via its tRNA synthatase-like domain [24–26] and phosphorylates serine-51 of the alpha subunit of translation initiation factor eIF2 (eIF2α). This leads to inhibition of general translation while promoting the translation of *GCN4* mRNA, by a mechanism dependent on its upstream open reading frames (uORFs) [27], which encodes a master transcription factor that positively controls amino acid biosynthesis genes.

Other conditions known to elicit the classic GAAC response (in *S. cerevisiae*) that may be relevant to this report are growth media that contain amino acid imbalances [28, see 29], and when specific amino acids provide the nitrogen source [30]. Also, hypomodified tRNAs undergoing degradation by the cytoplasmic rapid tRNA decay (RTD) pathway activate the GAAC response in a Gcn2-dependent manner, which was demonstrated in *S. cerevisiae* and *S. pombe* [31].

Similar to nuclear surveillance, other mechanisms can selectively promote *GCN4* mRNA translation activating a GAAC response in a Gcn2-independent manner. For example, Gcd⁻ mutants deficient in a translation initiation factor component in which GAAC is constitutively induced [29]. Mutants in certain tRNA modification activities also induce Gcn4-dependent GAAC, independent of Gcn2 [32, 33] [see 34].

A Gcd⁻ mutation in Trm61 of the Trm6/Trm61 tRNA methyltransferase required for modification of $m^1A58$ leads to destabilization of pre-tRNAi$^{Met}$ and its nuclear surveillance-mediated decay, resulting in GAAC activation independent of Gcn2 [17, 23]. Nuclear surveillance-induced GAAC in *S. cerevisiae* also occurs by expression of mutant tRNA genes or factors that cause nuclear pre-tRNAs to accumulate, and in such cases can be reversed by expression of tRNA maturation factors RNase P, La protein, modification enzymes, or the tRNA nuclear exporter, Los1 [22]. Likewise, deletion of *los1*Δ itself activates a GAAC response, with no apparent inhibition of general translation, thus appears to be Gcn2-independent [35].

Pre-tRNA degradation by nuclear surveillance has been widely conserved [15, 36–38]. As noted above, while multiple lines of evidence indicate a signalling branch of nuclear surveillance elicits GAAC response independent of Gcn2 in *S. cerevisiae*, to our knowledge this has not been established in another organism.

Coregulation of amino acids, tRNAs and broader metabolism exists although with species-specificity, a germane example being the GAAC responses in *S. pombe* and *S. cerevisiae* [39–42]. 3-aminotriazol (3AT) inhibits histidine synthesis thereby causing amino acid deficiency and Gcn4 activation in *S. cerevisiae* [43, 44] and also in human cells with ATF4 as the human Gcn4 homolog [45] [see 29]. This response is different in *S. pombe*, mediated by Fil1, which is non-orthologous but functionally akin to Gcn4 including in mode of translational regulation by uORFs [46, 47]. Also, *S. cerevisiae* contains one protein kinase, Gcn2 that phosphorylates eIF2α serine-51, while *S. pombe* and humans both contain more than one [reviewed in 46]. Specifically, *S. pombe* Gcn2 phosphorylates eIF2α in response to amino acid depletion whereas Hri1 and Hri2 differentially do so as part of responses to nitrogen and glucose starvation, and oxidative stress [48].

3AT induces many more than the GAAC genes in *S. cerevisiae* including in *gcn4*-mutant cells [29, 43]. 3AT also induces many genes in *S. pombe*, the GAAC genes via Fil1 in addition to genes that comprise the "core environmental stress response" (CESR) that can also be induced by specific stressors, hydrogen peroxide, the heavy metal cadmium, DNA-alkylating agents, osmotic stress and heat shock. Transcription factors Atf1 and Pcr1, neither of which are found in *S. cerevisiae* induce >100 common CESR genes, with a smaller number specific to each pathway, which differ in the extent to which Atf1 (and presumably its binding partner Pcr1) is engaged [49]. Relevant here is that although 3AT induces many genes, refined genetic analyses has led to the separation of 3AT-GAAC and 3AT-CESR gene "clusters" [see 46 and below].

*S. pombe* lacking La protein (*sla1Δ*) accumulate aberrant pre-tRNAs in rich (YES) or in minimal media (EMM) that differ only in the nitrogen source proline or NH4⁺ but grow very slowly in NH4⁺ [50]. For example, *sla1Δ* and *sla1*⁺ grow equally well in YES whereas addition of NH4⁺ to YES severely inhibits *sla1Δ* but not *sla1*⁺ [50]. It had been known that auxotrophy for leucine is exacerbated by NH4⁺ in mutants that antagonize target of rapamycin (TOR) signaling such as *tsc1Δ*, *tsc2Δ* and the *tor1Δ* mutant itself reflecting that *tor1*⁺ promotes leucine uptake [51, 52]. The *sla1Δ* mutant is more deficienct in leucine uptake in NH4⁺ media than a *tor1Δ* mutant in the same genetic background and this is a principal characteristic of its slow growth in EMM-NH4⁺ [50]. Although leucine uptake is also impaired in *sla1Δ* relative to *sla1*⁺ in EMM-Pro media, it is not as severe as in EMM-NH4⁺ and does not limit growth relative to *sla1*⁺ [50]. Thus while proline is a very poor nitrogen source relative to NH4⁺ which is excellent for growth of wild type *S. pombe* [51, 53], that *sla1Δ* cells grow comparably to *sla1*⁺ in proline but much slower than *sla1*⁺ in NH4⁺ illustrate the effects of Sla1 [50]. Thus *sla1Δ* and other mutants that disrupt tRNA biogenesis, modification and/or metabolism lead to rapamycin sensitivity and appear to antagonize TOR signalling [50, 54–56] [reviewed in 39], whereas deletion of *gaf1*⁺ which encodes the nitrogen-regulated GATA transcription factor that *represses* tRNA production, is resistant to TOR inhibition [52, 57]. Based on knowledge of the *S. cerevisiae* system, deficiency of amino acid uptake, aberrant tRNA processing or both might lead to GAAC activation in *sla1Δ* cells, by the classic pathway or in a Gcn2-independent manner.

We analyzed transcriptomes of *sla1Δ* cells including by comparison to a wealth of published data on other mutants. Genes upregulated in *sla1Δ* grown in media without NH4⁺ appear to reveal a GAAC response to altered tRNA homeostasis, whereas additional distinct genes upregulated in NH4⁺ media reflect a CESR. Deletion of the Rrp6 subunit of the nuclear exosome

decreases expression of GAAC genes by *sla1*Δ cells, providing evidence of a signalling component of nuclear surveillance in *S. pombe*. Spontaneous revertants (SSR) of *sla1*Δ slow growth in NH4⁺ were isolated in which the rapamycin hypersensitivity and GAAC expression phenotypes were also reversed. One mutant phenotype was characterized and shown to be due to a F32V mutation in Any1, an amino acid transport regulator previously linked to TOR signalng [58].

## Results

We previously reported overlap between the *sla1*Δ upregulated genes and a set of amino acid metabolism (AAM) genes that had been informatically grouped on the basis of a short upstream sequence motif thought to represent a transcription factor binding site [50]. Statistical significance of the overlap ranged from p< 4.75e$^{-17}$ to 3e$^{-12}$ in different growth media [50].

### *sla1*Δ cells upregulate a core set of genes as well as growth media-specific transcripts

As alluded to in the Introduction, *S. pombe* transcriptomes based on relevant biological results recently became available. Our transcriptomes analyzed here differed in abundance in *sla1*Δ (*h⁻ ade6-704 leu1-32 sla1*Δ*::ura4⁻*) and *sla1*⁺ cells (*h⁻ ade6-704 leu1-32 ura4⁻*) by 1.5x or more in each of both duplicate experiment samples after growth in three different media, EMM (Edinburgh minimal media) which contains NH4⁺ as the nitrogen source, EMM-Pro in which proline replaces NH4⁺, and YES (yeast extract with supplements). The analysis revealed 4, 9 and 11 down-regulated as well as 40, 60 and 64 upregulated genes in *sla1*Δ relative to *sla1*⁺ cells in YES, EMM-Pro and EMM-NH4⁺ media respectively. Thus by the number of genes whose expression increased or decreased by ≥1.5x, growth in YES had least affect (44 genes) whereas growth in EMM-NH4⁺ had greatest affect (75 genes) on the *sla1*Δ transcriptome.

The left panel of Fig 1A shows a three-way Venn graphic of the overlap of upregulated genes shared by *sla1*Δ cells grown in the three media. It reveals that 23 of the 40 *sla1*Δ genes upregulated in YES (58%), were also upregulated in both EMM-Pro and EMM-NH4⁺. Besides the 23 YES upregulated genes only 2 other were shared with the EMM-NH4⁺ set and 4 with the EMM-Pro set (Fig 1A left). Each of the two-way comparison Venn graphics include a statistical significance and representation factor (RF); a RF of >1 indicates more overlap than expected of two independent groups. Consistent with the three-way Venn, overlap of the EMM-Pro and EMM_NH4⁺ genes was the most highly significant, p < 3.349e$^{-64}$ with a RF = 54. The EMM-Pro and YES overlap was less significant, p < 1.16e$^{-46}$ but with higher RF = 62. The NH4⁺ and YES genes overlapped at p < 7.35e$^{-41}$ with higher RF = 53.7 (Fig 1A).

The 38 upregulated genes common to the EMM-Pro and EMM-NH4⁺ sets, and the 23 upregulated genes in all three sets, are hereafter referred to as "*sla1*Δ-com" and "*sla1*Δ-all." The identities of the genes in these sets are in, S1 Table. The *sla1*Δ-com gene set and the *sla1*Δ-all gene set were each subjected to gene ontology (GO) analysis using PomBase-defined GO terms [59, 60]. The 38 *sla1*Δ-com genes identified arginine, glutamine, and amino acid metabolic and biosynthetic processes represented by 8 distinct genes as well as two DNA-related processes (S2A Table). The DNA integration and recombination processes reflect transcripts from 8 *Tf2* retrotransposon loci each counting as a gene in this analysis as well as the *tlh1*⁺ DNA recQ helicase-like gene. *Tf2* retrotransposons from multiple loci are known to be upregulated in *S. pombe* GAAC-gene sets [46, 47] and in *tor*-deficient, in *tsc*-deficient mutants, and in mutants deficient in the retrotransposon chromatin repressor, *atf1*⁺ [61, 62].

The GO results reflect that the *sla1*Δ-com genes upregulated both in EMM with NH4⁺ or Pro include the *S. cerevisiae* homologs (in parantheses as designated in PomBase): *aes1*⁺

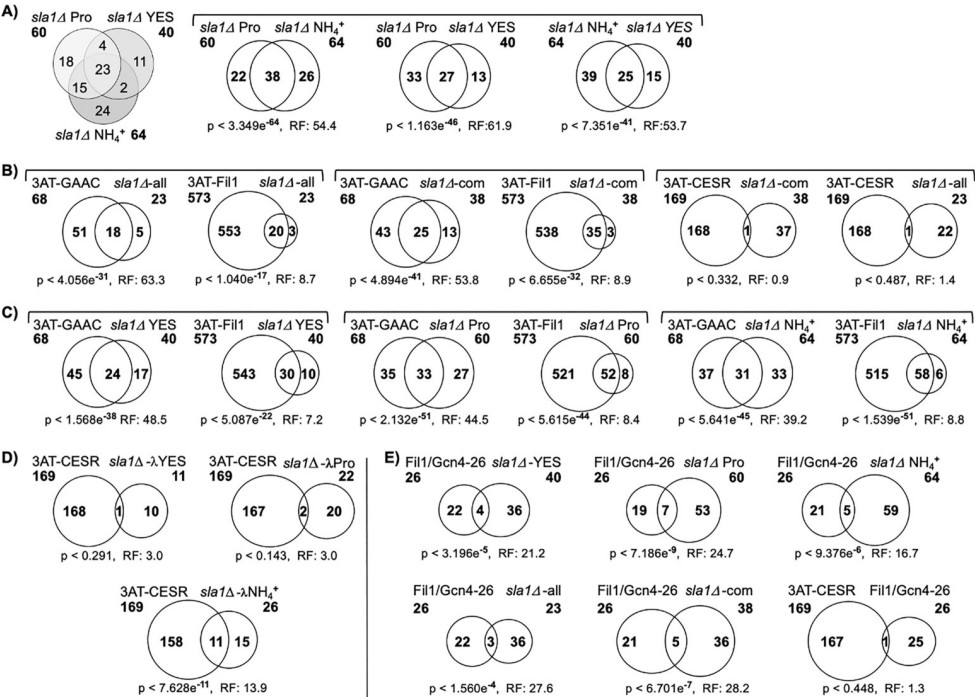

**Fig 1. Statistical significance of overlap of *sla1Δ* ≥1.5x transcriptomes and the representation factors (RF) (see text). A-E)** The representation factor (RF) is the number of overlapping genes divided by the expected number of overlapping genes drawn from two independent groups. RF > 1 indicates more overlap than expected of two independent groups and a RF < 1 indicates less overlap than expected. 3AT-GAAC and 3AT-CESR are from [46] in which these gene groups are referred to as clusters 3 and 2 respectively, and are considered analogous to a Gcn2-dependent GAAC response and a CESR response respectively. 3AT-Fil1 represents genes induced in response to 3AT in *gcn2*+ vs *gcn2Δ* cells [47]. Fil1/Gcn4-26 represents a core group of 26 genes directly bound by the Fil1 and Gcn4 transcriptional activators of GAAC genes in their respective genomes [47]. The *sla1Δ* Pro, *sla1Δ* YES and *sla1Δ* NH4+ gene sets are described in the main text, as are the *sla1Δ*-all and *sla1Δ*-com sets, and the *sla1Δ*-λPro, *sla1Δ*-λYES and *sla1Δ*-λNH4+ gene sets. Bold font numerals under these names indicate gene number in the set.

(*YHI9*), *arg3*+ (*ARG3*), *arg5*+ (*CPA1*), *arg12*+ (*ARG1*), *asn1*+ (*ASN1*), *gdh2*+ (*GDH2*), *pas1*+ (*PCL5*), SPAC56E4.03 (*ARO8*) and *aca1*+, involved in arginine, asparagine, glutamine, and glutamic acid metabolism (S2A Table). Six of these were upregulated in the *sla1Δ*-all gene set (S2A Table). GO analysis of the 23 genes in the *sla1Δ*-all set revealed multiple of the same processes involved in arginine, asparagine, glutamine, and glutamic acid metabolism as well as DNA processes (S2B Table). Thus, amino acid metabolic genes recognized as involved in cohesive processes are constitutively over-expressed in *sla1Δ* cells relative to *sla1*+ cells in all three growth media. Among the 15 upregulated genes shared by *sla1Δ* cells grown in EMM (but not YES), were *arg3*+, *arg5*+, *asn1*+, plus two involved in membrane transport, and others involved in related small molecule metabolism (S1 Table).

We also performed GO analysis on genes in the *sla1Δ* sets that do not overlap (S3A–S3C Table). The 11 genes in the YES set that do not overlap with either of the two other sets were designated as *sla1Δ*-λYES. The 22 and 26 genes in the *sla1Δ* Pro and NH4+ sets that do not overlap with eachother were designated *sla1Δ*-λPro and *sla1Δ*-λNH4. Interestingly, 3 of the 11 genes specific to *sla1Δ*-λYES and 16 of the 22 genes specific to *sla1Δ*-λPro were linked to multiple GO related metabolic processes at p <0.01 (S3A and S3B Table). By contrast, the 26 genes specific to *sla1Δ*-λNH4 yielded *no* GO terms at p <0.01 (not shown); only when the stringency was lowered to p <0.05 was a single GO term revealed (S3C Table).

In conclusion, a larger number of genes were linked to nutrient transport and amino acid metabolism in *sla1*Δ Pro genes than the other two corresponding gene sets: 30 genes in the *sla1*Δ Pro set of 60 (50%), 13 genes in the YES set of 40 genes (33%), and 9 genes in the *sla1*Δ NH4⁺ set of 64 (14%) (S4–S6 Tables). Thus a significant fraction of upregulated genes in *sla1*Δ cells are shared in all three media and involved in biosynthesis and metabolism of the basic nitrogenous amino acids (S2B Table), while a number of other genes are upregulated to different extents. The GO analysis shows that the *sla1*Δ YES and EMM-Pro genes appear more similar to each other than to the EMM-NH4⁺ genes in their links to amino acid metabolism. This is especially notable because YES is rich media and EMM is minimal media with NH4⁺ considered to be an excellent nitrogen source for *S. pombe* growth while proline is a very poor nitrogen source by comparison [51, 53]. This issue with regard to the *sla1*Δ cells is addressed in a later section.

## *sla1*Δ cells produce different GAAC-like responses in different growth media

The analyses indicate that *sla1*Δ cells grown in EMM-Pro or EMM-NH4⁺, defined media that differ only in the nitrogen source, upregulate a significant number of the same genes as well as different genes. Also, growth of *sla1*Δ cells is specifically sensitive to the inhibitory effects of NH4⁺ relative to isogenic *sla1*⁺ cells, whereas they grow equally well in EMM-Pro [50]. We examined the *sla1*Δ upregulated gene sets by comparing them to available sets of *S. pombe* stress response genes [46, 47]. Gene deletions were used to dissect *S. pombe* responses to 3AT and led Udagawa et al. to derive a "Cluster 3" subset of 68 upregulated genes that is considered analogous to the Gcn2-dependent GAAC response in *S. cerevisiae*, and a "Cluster 2" subset of 169 upregulated genes that represents the CESR [46]. Duncan et al. later identified and characterized Fil1, a *S. pombe* transcription factor whose mRNA responds to 3AT-induced signalling analogously to *S. cerevisiae* Gcn4; the translated Fil1 protein then activates *S. pombe* GAAC (and other) genes in a manner similar to Gcn4 (below) [47]. Duncan et al. isolated a set of 573 genes induced in response to 3AT in *gcn2*⁺ but not in *gcn2*Δ cells [47]. These *S. pombe* gene sets are referred to as 3AT-GAAC, 3AT-CESR and 3AT-Fil1 in Fig 1B and hereafter.

The *sla1*Δ-all set of 23 genes significantly overlapped with each of the 3AT-GAAC and 3AT-Fil1 sets, as did the *sla1*Δ-com set of 38 genes (Fig 1B, left and middle). In contrast to the significant overlaps with these 3AT gene response genes, there was no significant overlap of *sla1*Δ-com nor *sla1*Δ-all genes with the 3AT-CESR (Fig 1B, right). Comparison of the full upregulated gene sets of *sla1*Δ YES, EMM-NH4⁺ and EMM-Pro to the GAAC and Fil1 sets revealed significant overlaps, each with distinctive RFs (Fig 1C).

The *sla1*Δ gene sets were distinguished by comparison to 3AT-CESR genes. This used *sla1*Δ genes from each set not shared with one or both other sets (Fig 1A left, S3A–S3C Table): *sla1*Δ-λYES, *sla1*Δ-λPro and *sla1*Δ-λ NH4⁺ (Fig 1D). By comparison to the *sla1*Δ-λYES and the *sla1*Δ-λPro genes which did not exhibit significant overlap with 3AT-CESR genes (p < 0.291 & p < 0.143), the *sla1*Δ-λ NH4⁺ genes exhibited significant overlap (p < 7.628e⁻¹¹) and with higher RF (Fig 1D).

By analyzing ChIP-seq data Duncan et al. defined a core group of 26 genes that are directly bound by the Fil1 and Gcn4 transcriptional activators of GAAC genes in their respective genomes [47]. Comparison to this "Fil1/Gcn4-26" data set indicates somewhat higher overlap with the *sla1*Δ EMM-Pro genes than with *sla1*Δ YES and with *sla1*Δ EMM-NH4⁺ (Fig 1E, top row). All of the *sla1*Δ genes in these sets that overlap with the Fil1/Gcn4-26 set are involved in amino acid biosynthesis/metabolism (S7 Table).

Fig 1 and the GO analyses results in S2–S5 Tables suggest that cells deleted of *sla1*⁺ that are grown in all three growth media exhibit constitutive upregulation of a set of GAAC response

genes albeit to slightly different extents. However *sla1Δ* cells grown in EMM-NH4⁺ also upregulate a number of genes distinctive to the CESR as reflected by Fig 1D, in addition to the GAAC genes. The *hsp3101*⁺, *hsp3102*⁺, *isp6*⁺, *pcr1*⁺ and *zym1*⁺ transcripts which are known to be induced under multiple stresses [46, 49] were found in the upregulated *sla1Δ* cell transcripts specifically in the EMM-NH4⁺ set (S1 Table).

## Further evidence of composite input to the GAAC response in *sla1Δ* cells

Sla1 is known to protect pre-tRNA from a nuclear surveillance-mediated decay pathway that includes pre-tRNA polyadenylation and is dependent on *rrp6*⁺ of the nuclear exosome [20]. Rrp6 deletion rescues a nuclear surveillance-mediated GAAC-related growth phenotype of *S. cerevisiae* that occurs in response to m$^1$A58 hypomodification of pre-tRNAi$^{Met}$ as described above [23]. Deletion of *rrp6*⁺ significantly compromises *S. pombe* growth under multiple conditions, including in rich media [20, 63–65], however *RRP6* deletion from *S. cerevisiae* has little effect on growth [66, 67] other than at 37˚C [68]. Our objective for Fig 2 was to ask if nuclear surveillance may be involved in signaling GAAC gene induction in *sla1Δ* cells a similar way by *rrp6*⁺ deletion in *S. pombe*. Examination by northern blotting shows that *sla1Δ* cells exhibit increased levels of *gdh2*⁺ mRNA (C312.04 in ref [50]) whereas *sla1*⁺ *rrp6*⁺ (WT) cells and *sla1Δ* cells deleted for *rrp6*⁺ (*rrp6Δ*) have lower levels (Fig 2A). For this we also examined *lys4*⁺ levels (Fig 2A) because it is in the 3AT-GAAC gene set [46] the 3AT-Fil1 set and the Fil1/Gcn4-26 set [47] (LYS21/LYS20 in *S. cerevisiae*, S7 Table) but was not included in the *sla1Δ* transcriptome sets because it did not meet criteria of detection at ≥1.5x relative to *sla1*⁺ on each of the duplicate experiments; it was detected at 2.1x and 1.46x in YES media and at 2.40x and 1.43x in EMM-NH4⁺ (not shown), and this was consistent with our northern blot results (Fig 2B, see Y-axes). The ribosomal protein *rpl8*⁺ mRNA served as a loading control and was used for normalization of *gdh2*⁺, *aca1*⁺ and *lys4*⁺ mRNAs (Fig 2A and 2B).

As nothern blotting is not as direct nor standardized an approach for mRNA quantification as is gene array methodology, some comments are noteworthy. First, visual inspection of Fig 2A confirms that the *gdh2*⁺ and *aca1*⁺ GAAC mRNAs examined, are reproducibly and robustly upregulated in *sla1Δ* relative to *sla1*⁺ WT cells. Second, because quantification of these mRNAs in Fig 2B uses *rpl8*⁺ for normalization and reports levels relative to WT, it is important to note that actual signal intensity of probe detection on the blot panels depicted in Fig 2A were indeed nearly 4-fold and 3-fold higher for *gdh2*⁺ and *aca1*⁺ mRNAs in *sla1Δ* cells in EMM-NH4⁺ media as compared to *sla1Δ* cells in YES.

The analysis showed that *aca1*⁺, *gdh2*⁺ and *lys4*⁺ levels were lower in the *sla1Δ rrp6Δ* double mutant than in *sla1Δ* indicating that deletion of *rrp6*⁺ suppressed their expression (Fig 2A and 2B). Finally, the cumulative quantitative northern blot data were tested for statistical sigificance (Fig 2C). Most relevant this showed that the differences between *sla1Δ* cells and *sla1Δ rrp6Δ* cells in normalized mRNA levels for *aca1*⁺ and *gdh2*⁺ were significant ($p = 0.0089$) supporting the visual evidence that deletion of *rrp6*⁺ from *sla1Δ* suppresses activation of *aca1*⁺ and *gdh2*⁺ GAAC gene expression.

The *sla1Δ* cells grow comparably to *sla1*⁺ in YES but slower than *sla1*⁺ in EMM-NH4⁺ [50] (Fig 2C). The effects of *rrp6*⁺ deletion relative to *sla1Δ* was media-dependent. The *rrp6Δ* cells exhibited reduced growth relative to WT and *sla1Δ* on YES while they grew better than *sla1Δ* on EMM-NH4⁺ (Fig 2C). Growth of the *sla1Δ rrp6Δ* double mutant was poor relative to either of the single mutants on YES, and was markedly worse than the *rrp6Δ* single mutant on EMM-NH4⁺ (Fig 2C). Thus, the results were clear that while *rrp6*⁺ deletion could rescue upregulated GAAC gene expression in *sla1Δ rrp6Δ* cells, it could not rescue the growth deficiency associated with *sla1Δ* cells.

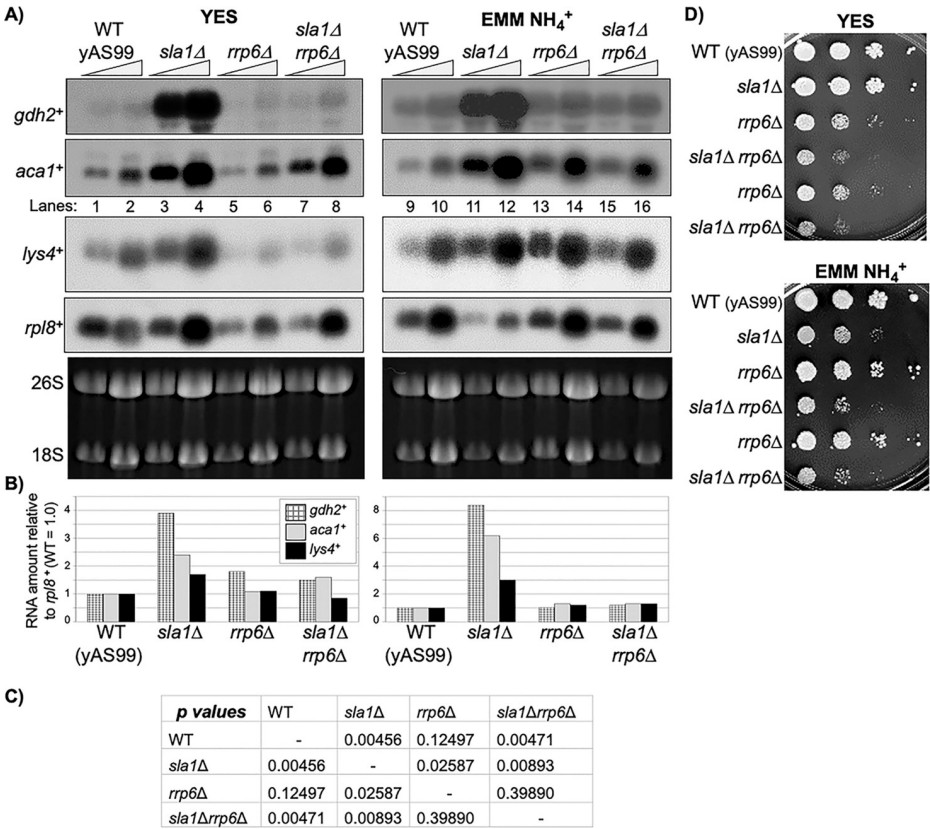

**Fig 2. Evidence that the Rrp6 subunit of the nuclear exosome contributes to GAAC gene expression in *sla1Δ* cells.**
**A)** Northern blot analysis of RNA from cells grown exponentially in rich medium (YES) (left) or Edinburgh minimal medium (EMM) containing NH4⁺ (right). Total RNA, 5 and 10 μg of each sample were separated on a 1% denaturing agarose gel, transferred to a GeneScreen Plus nylon membrane, and subjected to sequential hybridization with ³²P-labelled probes for mRNAs *ghd2*⁺, *aca1*⁺, *lys4*⁺ and *rpl8*⁺ the latter of which serves as a quantitative normalization control, as indicated. A photo of the ethidium bromide stained agarose gel prior to transfer is shown as the bottom panel with the 18S and 26S rRNA bands indicated. **B)** Quantitation of the *ghd2*⁺, *aca1*⁺ and *lys4*⁺ bands relative to the *rpl8*⁺ mRNA that encodes ribosomal protein L8. Values represented by each bar indicate two separate RNA sample loadings per experiment (technical duplicates), relative to WT which was set to 1. **C)** The quantitative data reported for *ghd2*⁺ and *aca1*⁺ mRNA levels in both panels of B were used to derive *p*-values using a basic T-test of statistical significance of the differences indicated. **D)** Cells were grown in liquid medium and serial dilutions were spotted to the indicated plates and incubated at 32°C for 2 to 6 days. Strains: yAS99 (WT); yAS113 (*sla1Δ*); yYH7a (*rrp6Δ*); CY1670 (*sla1Δ rrp6Δ*).

Rrp6 deletion rescues a nuclear surveillance phenotype of *S. cerevisiae*, that is due to degradation of a single pre-tRNAi^**Met** species owing to its unique sensitivity to m¹A58 hypomodification in the *trm6-504* mutant [23]. Fig 2 shows that suppression of GAAC gene upregulation in *sla1Δ rrp6Δ* cells is apparently uncoupled from the slow growth phenotype of *sla1Δ rrp6Δ*. As *sla1Δ* cells slow their growth specifically in NH4⁺ media we attempted to uncover negative regulators of this phenotype.

## Isolation of *sla1Δ* spontaneous revertants (SSR) of slow growth in NH4⁺ media

*sla1Δ* cells plated on EMM-NH4⁺ led to the appearance of very rare relatively fast growing colonies that were reproducibly surrounded by smaller sattelite colonies (Fig 3A). After plating 125,000 cells, a rare colony arose and satellite colonies appeared thereafter as in Fig 3B. The fast growth colonies isolated after a single plating were named *sla1Δ* spontaneous revertants

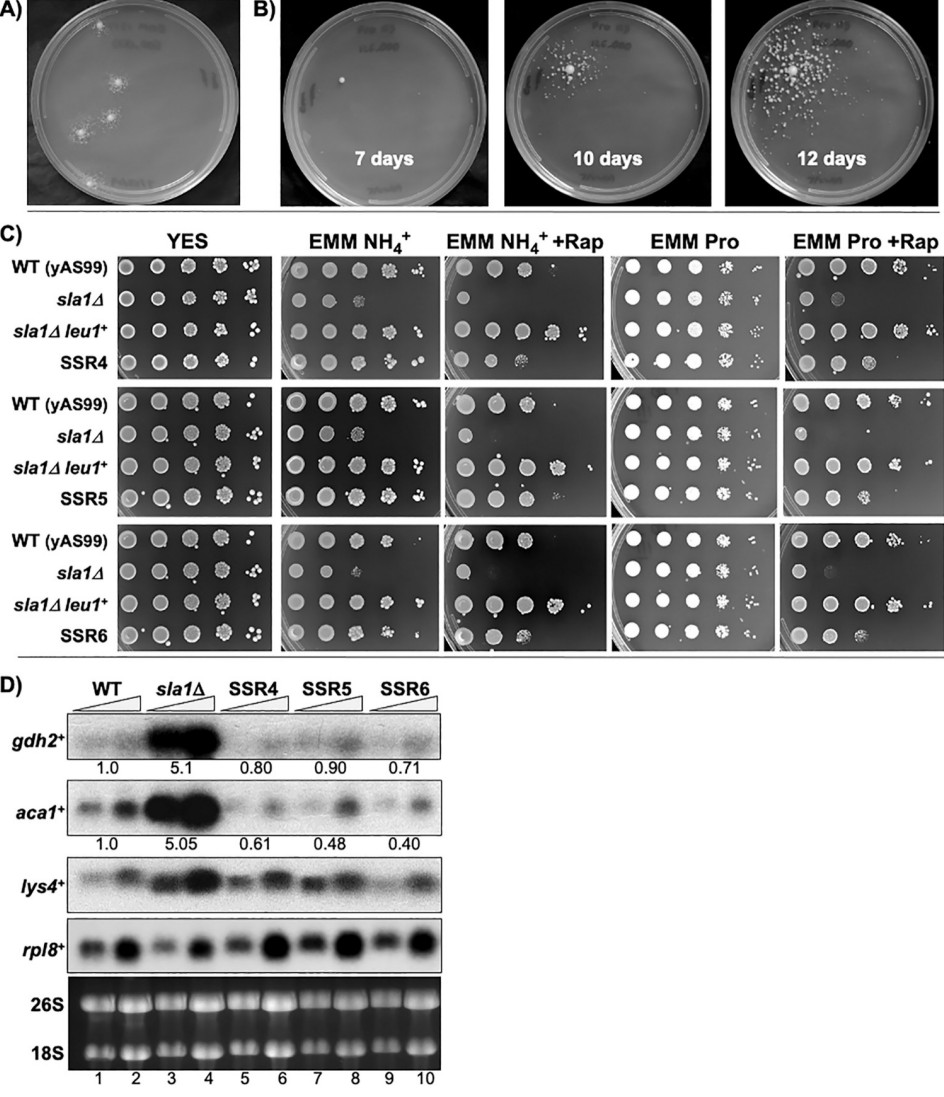

**Fig 3. Characterization of spontaneous *sla1Δ* slow growth revertants (SSRs). A)** Results of plating *sla1Δ* cells (yAS113) on EMM-NH4+. **B)** 250,000 *sla1Δ* cells plated and photographed after incubation for 7, 10 and 12 days. **C)** WT (yAS99) *sla1Δ* (yAS113), and a *leu1+* transformant of yAS113, *sla1Δ leu1+*, is shown alongside SSR4, SSR5 and SSR6 in the three sets. Rapamycin was used at 50 ng/ml. **D)** Northern blot of 5 and 10 μg total RNA from the strains indicated above the lanes grown in EMM-NH4+ probed for *ghd2+*, *aca1+*, *lys4+* and *rpl8+* mRNAs as in Fig 2. Numbers under the panels indicate the average of duplicate quantification measurements of the bands relative to *rpl8+* which serves as quantitative normalization relative to WT which was set to 1 as in Fig 2.

(SSRs) followed by a number. These were streaked and single colony derivatives of each were screened by comparing their growth to WT, *sla1Δ* and small sattelite colonies, by serial dilution spotting on YES, EMM-NH4+ and EMM-Pro. Additional analyses are shown in Fig 3C.

Fig 3C shows that SSRs 4, 5 and 6 share a similar pattern of media and rapamycin sensitivities that is distinct from both the WT and *sla1Δ* strains. As noted, auxotrophy for leucine in the presence of NH4+ in the media elicits sensitivity to rapamycin (Rap) in *S. pombe* mutants in which TOR signalling is antagonized [51, 69]. Use of *leu1-32* auxotrophs as "WT" *S. pombe* strains is not uncommon ([70], e.g., strain FY1862 in [71–73] and SPG17 in [74, 75]). As reported, deletion of *sla1+* from the leucine auxotrophs yAS99 and SPG17, greatly exacerbates the NH4+ dependent Rap sensitivity [50]. Reintegration of *leu1+* at its native locus as in *sla1Δ*

*leu1*+, suppressed the slow growth in EMM-NH4+ and led to increased growth in EMM-NH4+ +Rap relative to the *leu1-32 sla1*+ WT strain yAS99 (Fig 3C). Notably however, *sla1Δ leu1*+ did not show growth advantage over yAS99 in EMM-Pro+Rap.

The slow growth of *sla1Δ* in EMM-NH4+ and EMM-NH4++Rap was reversed in the SSRs with a pattern that suggests suppression of leucine auxotrophy although not as completely as by restoring *leu1*+. The SSRs grew much better than *sla1Δ* and similarly to WT in all media except EMM-Pro+Rap (Fig 3C). The relative slow growth of the SSRs in EMM-Pro+Rap as compared to EMM-NH4++Rap may likely reflect the differential effects of these nitrogen sources on the regulation of amino acid permeases [51]. This comparative analysis reveals that the SSRs are distinct relative to WT and *sla1Δ*. A northern blot in Fig 3D shows that mRNAs from representative GAAC genes *gdh2*+ *aca1*+ and *lys4*+ in *sla1Δ* cells, are robustly suppressed in the SSRs as seen in WT.

### Sequence identification of SSR mutations

We previously used whole genome sequencing to identify mutations in spontaneous mutants that exhibited a phenotype after one plating [76]. Genome sequencing of SSR4, SSR5 and SSR6 was performed using the parent *sla1Δ* strain whole genome sequence as a reference genome. Single nucleotide polymorphisms (SNPs), insertions, deletions and copy number were scrutinized as described [76]. For SSR5, two gene-associated SNPs were identified: one that changes a UUC to a GUC codon (F32V) in a gene named *any1*+ (arrestin in nutrient response in yeast-1, SPBC18H10.20c) that was reported to control amino acid uptake [58, 77], and another in the 3' UTR of *eki1*+ (SPAC13G7.12c) a choline/ethanolamine kinase. For SSR6, a four base pair deletion was identified on chr II:3487055–3487059 that would interfere with a transcription start site for *zfs1*+ (SPBC1718.07c) annotated as selectively active during nitrogen starvation [78]. *zfs1*+ is known to regulate the G1 cyclin *puc1*+ in response to nitrogen depletion or inhibition of TOR signaling [79]. No high confidence mutations could be found in SSR4.

With 16 promoters and many more transcription start sites, *zfs1*+ has more promoters than the great majority of *S. pombe* genes [80] and was not examined further. We focused on the *any1*-F32V missense mutation in SSR5. *any1*+ encodes an arrestin-related endocytic adaptor that regulates trafficking of amino acid plasma membrane permeases *aat1*+ (leucine and other aa uptake, *GAP1* in *S. cerevisiae*) and *cat1*+ (arginine/lysine uptake, *CAN1* and others in *S. cerevisiae*); *any1*+ is known to suppress defects in amino acid uptake due to mislocalization of plasma membrane permeases associated with nutritional stress [58, 77]. Any1 binds more stably to its E3 ubiquitin ligase partner Pub1 (Rsp5 in *S. cerevisiae*) in states of hypofunction and limits Aat1 function at the plamsma membrane [77]. Moreover, proper TOR activity is required for Aat1 trafficking to the plasma membrane during nutrient stress [81]. These results are consistent with the SSR5 phenotype and the possibility that *any1*-F32V has decreased ability to limit Aat1 leucine uptake function in EMM-NH4+ media.

### *any1*-deletion suppresses slow growth of *sla1Δ* in EMM-NH4+

Deletion of *any1*+ from WT yAS99 and from *sla1Δ* reduced their growth in YES, but had little effect on WT growth in EMM-NH4+ and rescued the growth deficiency of *sla1Δ* in EMM-NH4+ (Fig 4A). Thus, growth of *any1Δ* is differentially sensitive to YES and EMM-NH4+, whereas *sla1Δ* cells exhibit the opposite sensitivity. Most important, deletion of *any1*+ largely rescued the slow growth of *sla1Δ* on EMM-NH4+, while the strains grew comparably on EMM-Pro (Fig 4A). These data are consistent with *any1*+ as the target gene whose mutation is responsible for the SSR5 growth phenotype and the idea that *any1*-F32V is a reduced function allele in these cells.

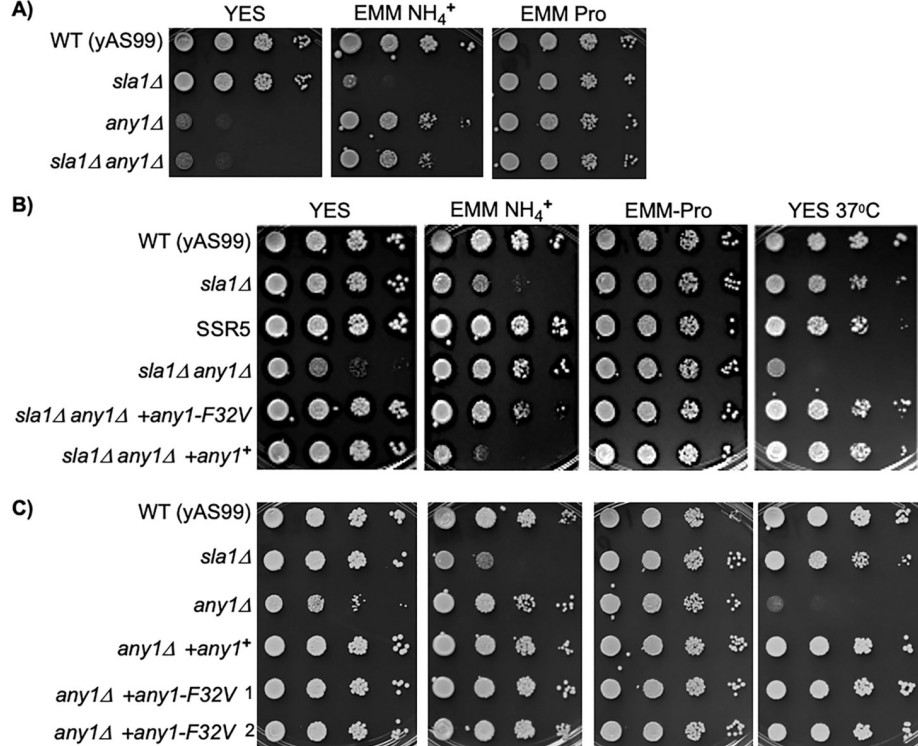

**Fig 4. *Any1-F32V* is a mutant allele that selectively suppresses slow growth of *sla1Δ*.** Cells grown in liquid medium were serially diluted and spotted onto the indicated plates and incubated at 32°C or 37°C as indicated, for 2d (panel A) to 6d (panels B & C). When comparing a strain in different panels note colony size (as well as cell density) as an independent indicator of relative growth, for example *any1Δ* vs. *sla1Δ* in panel 4°C YES. The strains are as follows. **A)** yAS99 (WT); yAS113 (*sla1Δ*); CY1698 (*any1Δ*); CY1697 (*sla1Δ any1Δ*). **B)** yAS99 (WT); yAS113 (*sla1Δ*); CY1665 (SSR5); CY1697 (*sla1Δ any1Δ*); CY1711 (*sla1Δ any1Δ +any1-F32V*); CY1710 (*sla1Δ any1Δ +any1+*). **C)** yAS99 (WT); yAS113 (*sla1Δ*); CY1698 (*any1Δ*); CY1712 (*any1Δ +any1+*); CY1713 (*any1Δ +any1-F32V*); 1 & 2 show two different isolates.

## Validation of *any1-F32V* as a hypofunctional allele that suppresses *sla1Δ* slow growth in EMM-NH4+

In order to assess potential effect of the F32V mutation, we integrated *any1+* and *any1-F32V* into the *sla1Δ any1Δ* double mutant and examined growth along with WT, *sla1Δ*, SSR5, and *any1Δ sla1Δ* (Fig 4B). Providing *sla1Δ any1Δ* with the *any1+* allele or the *any1-F32V* allele rescued growth on YES (*sla1Δ any1Δ +any1+*, and *sla1Δ any1Δ +any1-F32V*) (Fig 4B). This provided evidence that *any1-F32V* is not a null allele and more importantly showed that it confers growth advantage to *sla1Δ* on EMM-NH4+ media while the WT *any1+* does not (Fig 4B, *sla1Δ any1Δ +any1+*, and *sla1Δ any1Δ +any1-F32V*). This validated the *any1-F32V* allele mutation as largely responsible for reversing the slow growth of *sla1Δ* on EMM-NH4+.

It was previously reported that slow growth of *sla1Δ* cells reflect two deficiencies, low plating efficiency and slow proliferation, and that deletion of certain genes, *pub1+* (*sla1Δ pub1Δ*) suppresses one much more than the other (proliferation, small colony size) [50]. By limiting growth to relatively short times, Fig 4A suggests that a function of *any1+* is to support efficient growth in YES. Thus, we could assess the degree to which *any1-F32V* might be generally hypofunctional by assessing to what degree if any it could restore growth of the *any1Δ* strain on YES. On plates in Fig 4C which were allowed to grow for longer time than in panel A, the *any1Δ* strain was complemented with a chromosomal copy of either *any1+* or *any1-F32V*.

Both alleles rescued the slow growth of *any1Δ* on YES and on the other media, including YES at 37°C (Fig 4C). Thus *any1*⁺ and *any1-F32V* were indistinguishable for function in restoring growth in YES media and in EMM-NH4⁺ in the presence of *sla1*⁺ (Fig 4C, *any1Δ +any1*⁺ vs. *any1Δ +any1-F32V*) whereas *any1-F32V* differentially conferred growth advantage to *sla1Δ* in EMM-NH4⁺ but *any1*⁺ is defective for this activity in *sla1Δ* cells (Fig 4B, *sla1Δ any1Δ +any1*⁺ vs. *sla1Δ any1Δ +any1-F32V*). By this assay in Fig 4C, the ability to restore growth in YES, *any1*-F32V does not appear to be a simple hypomorphic allele. The cumulative data suggest that F32V is a distinctive mutation that alters *any1*⁺ function in *sla1Δ* cells to enhance growth in otherwise restrictive EMM-NH4⁺ media.

## Effects of *any1*⁺ and *any1-F32V* on GAAC gene expression

We performed northern blotting including examination of *any1*⁺ mRNA expression. The third panel of Fig 5 shows that *any1*⁺ mRNA was comparable in WT, *sla1Δ* and SSR5 (lanes 1–6), as expected for the F32V mutation acting via Any1 protein as opposed to affecting mRNA levels. This is supported by comparison to the *rpl8*⁺ panel used as a loading control.

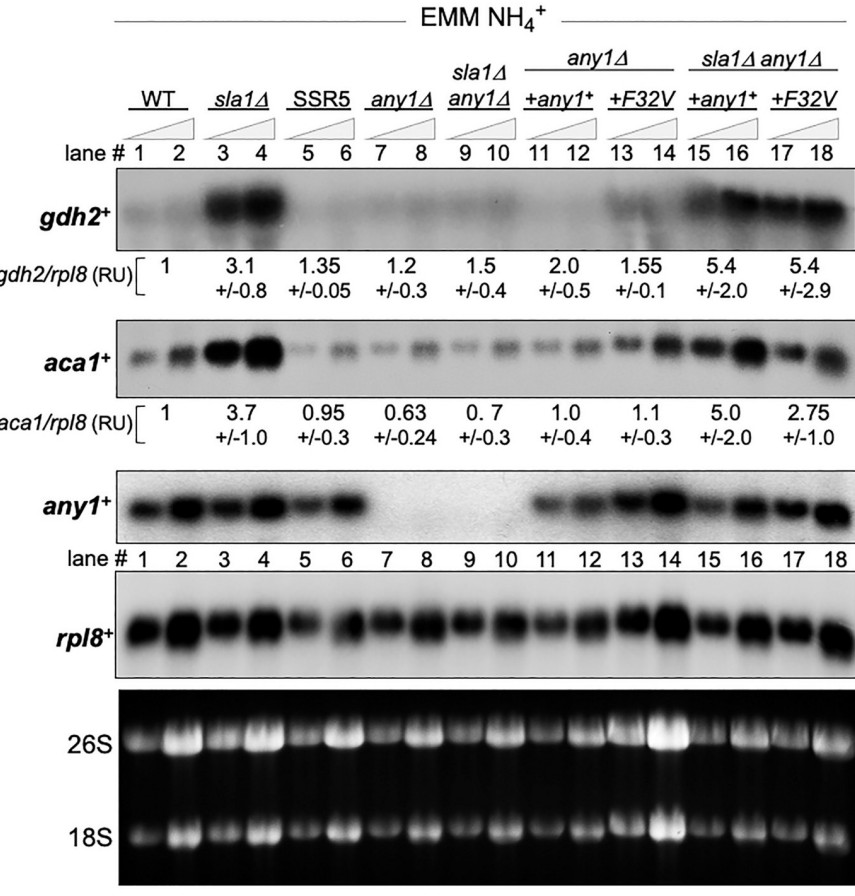

**Fig 5. Evidence that *any1-F32V* may selectively suppress *aca1*⁺ gene expression in *sla1Δ* cells.** A northern blot of 5 and 10 μg of total RNA from cells grown in EMM-NH4⁺, as in Fig 2. The strains are as described in Fig 4. The quantification values indicated under the lanes of the *gdh2*⁺ and *aca1*⁺ panels were derived from two biological experiments each containing measurements of two separate RNA sample loadings calibrated by the loading control *rpl8*⁺ mRNA in each lane and reported as relative units (RU). The lower panel shows the ethidium stained agarose gel prior to transfer.

Probing for *gdh2*+ mRNA showed that *any1Δ* cells express *gdh2*+ mRNA levels comparable to WT and SSR5, substantially lower than in *sla1Δ* (Fig 5, lanes 1–8). Quantification of the *gdh2*+ and *aca1*+ mRNA levels are shown under the lanes of those panels. Also important, deletion of *any1*+ from *sla1Δ* as in *sla1Δ any1Δ*, suppressed *gdh2*+ and *aca1*+ mRNA levels (compare lanes 3–4, 9–10).

We next examined effects of integrating *any1*+ and *any1-F32V* into the *any1Δ* strain and the double mutant *sla1Δ any1Δ* (Fig 5, lanes 11–18). Both alleles restored *any1*+ mRNA to the *any1Δ* and *sla1Δ any1Δ* strains at comparable levels (compare lanes 7–10 with 11–18 of lower three panels). The *any1-F32V* allele in lanes 17–18 failed to suppress *gdh2*+ and *aca1*+ levels to that in SSR5 (lanes 5–6). However, examination of the blots suggested that *any1-F32V* suppressed *aca1*+ expression more than it suppressed *gdh2*+ (lanes 15–18). Thus although *any1-F32V* failed to recapitulate suppression observed in SSR5, it appeared to suppress *aca1*+ more than *gdh2*+. More specifically, the *any1-F32V* allele appeared to lead to lower *aca1*+ levels than the *any1*+ allele after integration *into sla1Δ any1Δ* (lanes 15–16 vs. 17–18), whereas *gdh2*+ levels appeared more similar (lanes 15–16 vs. 17–18). However, more convincing is that integration of neither *any1-F32V* nor *any1*+ could recapitulate the suppression of *gdh2*+ and *aca1*+ observed in SSR5. We note that although *any1*+ and *any1-F32V* are associated with distinct functional activities in *sla1Δ* and SSR respectively, it is possible that *aca1*+/*gdh2*+ mRNA levels were not recapitulated by re-integation into the disrupted *any1*+ locus of the *any1Δ sla1Δ* mutant because the correct regulatory control regions were not properly re-established.

Are the observations regarding the apparent differences in *aca1*+ mRNA levels that were observed by inspection of Fig 5 relevent to growth in EMM-NH4+? Fig 4B shows that while *sla1Δ* vs. *sla1Δ any1Δ +any1*+ show comparable poor growth, whereas *sla1Δ any1Δ +any1-F32V* is intermediate, not as good as SSR5 but clearly better than *sla1Δ any1Δ +any1*+. This suggests that *any1-F32V* is a positive growth determinant in *sla1Δ*-SSR5 in EMM-NH4+, distinguished from *any1*+.

## SSR5 (*any1-F32V*) exhibits increased leucine uptake in EMM-NH4+ compared to *sla1Δ* cells

As noted, deficiency of leucine uptake in NH4+ media is likely the key basis of the slow growth of *sla1Δ* cells [50]. We therefore compared $^3$H-leucine uptake by *sla1Δ*, SSR5, *any1Δ* and related strains using the same assay as before [50, 51]. The data were consistent with previous results [50, 51] and showed that deficiency of leucine uptake of *sla1Δ* was clearly reversed in SSR5 (Fig 6). Moreover, *any1Δ* cells as well as *sla1Δ any1Δ* and SSR5 cells exhibited greater uptake than WT cells. The data fit with *any1*+ as a negative regulator of leucine uptake in the presence of NH4+ in *sla1Δ* cells [58, 82], and with *any1-F32V* as relieving this negative influence.

## Discussion

A principal feature of yeast that lack La protein is aberrant pre-tRNA processing [9, 20, 83–88]. Deletion of *sla1*+ from different WT laboratory strains leads to hypersensitivity to rapamycin which reverts to rapamycin resistance upon reintroduction of *sla1*+ [50]. Separately, *sla1Δ* and *sla1*+ cells grow comparably to each other in YES and comparably to each other in EMM-Pro, whereas *sla1Δ* grows much slower than *sla1*+ in EMM-NH4+ [50]. The first objective of this study was to compare the *sla1Δ* ≥1.5x transcriptomes in the three media, and to other relevant transcriptomes. Growth of *sla1Δ* in YES, EMM-Pro and EMM-NH4+ leads to 40, 60 and 64 upregulated genes at ≥1.5x relative to *sla1*+ cells. Such comparisons and GO analyses reveals that all three *sla1Δ* transcriptomes share 23 genes ('*sla1Δ*-all') of which 18 overlap with

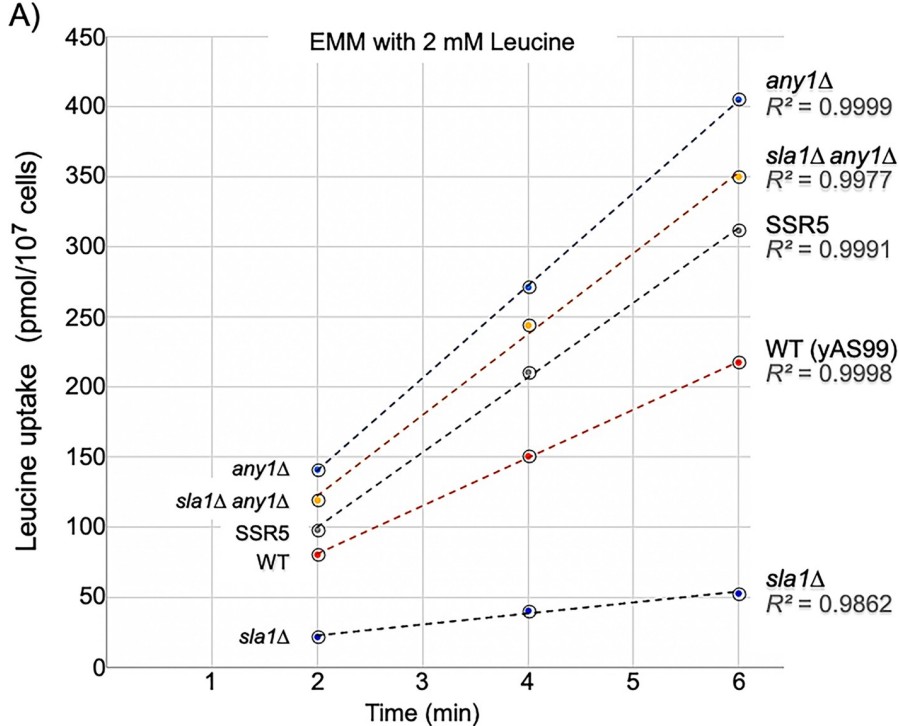

**Fig 6. SSR5 (*sla1Δ any1-F32V*) and *any1Δ* cells exhibit elevated leucine uptake relative to *sla1Δ*. A)** Cells were grown in EMM-NH4+, transferred to fresh EMM-NH4+ media with ³H-leucine, and assayed for ³H-leucine uptake at the time points indicated. The $R^2$ values for each of the trendlines representing the claculated slopes of the rates of leucine incorporation are indicated to the right. Strains: yAS99 (WT); yAS113 (*sla1Δ*); CY1665 (SSR5); CY1697 (*sla1Δ any1Δ*); CY1698 (*any1Δ*). **B)** Multiple regression/correlation analysis was used to derive *p*-values from the ³H-leucine uptake data (see methods).

the "cluster 3" 3AT-GAAC transcriptome [46] and 20 of which overlap with the 3AT-Fil1 transcriptome [47] (Fig 1A, 1B; S1, S2B Tables). Further analysis shows that the *sla1Δ* EMM-Pro and EMM-YES transcriptomes are 55–60% aligned with the 3AT-GAAC response and 2–3% with the 3AT-CESR, whereas <50% of the EMM-NH4+ transcriptome aligned with the 3AT-GAAC response and 17% with the CESR (Fig 1C and 1D). Although the NH4+ transcriptome is set apart from the other two, the genes specific to it reflect stress but do not satisfactorily explain the slow growth (below).

Programmed genetic responses in *S. pombe* and *S. cerevisiae* that include the GAAC (following exposure to 3AT) are complex, include several hundred genes in various functional groups, and reflect multiple transcription factor 'activities' [29, 43]. In *S. cerevisiae*, activation of GAAC genes can be triggered by uncharged tRNA following inhibition of amino acid synthesis [29, 89], imbalances of amino acids in the growth media and in particular when serving as the nitrogen source [28, 30], mischarged tRNAs [41], hypomodified m¹A58 pre-tRNAi^Met [17, 18, 23], hypomodifications of the tRNA body that lead to rapid tRNA decay [31], tRNA

anticodon hypomodifications that impair mRNA decoding [32], nuclear unprocessed aberrant pre-tRNAs that trigger nuclear surveillance [21, 22], and deletion of *LOS1*, loss of which leads to accumulation of intron-containing nuclear pre-tRNAs [35, 90].

## Differential and composite GAAC stress responses in *sla1Δ* cells in EMM-Pro and NH4[+]

Our analyses indicate that *sla1Δ* cells grown in EMM-YES and -Pro constitutively express a core genetic program that is in accord with the GAAC response more than any other response characterized in *S. pombe* [30, 46, 47, 49, 57, 91], the hallmark feature of which is upregulation of genes involved in amino acid biosynthesis and metabolism. To further analyze the *gcn2*[+] *and fil1*[+] dependent 3AT response in *S. pombe*, to segregate it from other components of the response and relate it to the classic 3AT response in *S. cerevisiae*, Duncan et al. identified a core set of 26 orthologous genes strongly enriched in amino acid biosynthesis functions, that are bound by the Gcn4 and Fil1 transcription factors [47]. It is highly significant (p< 7.186e[-9]) that 7 of these Fil/Gcn4-26 genes are in the *sla1Δ* EMM-Pro set of constitutively upregulated genes (Fig 1E), *arg5*[+] (CPA1), *arg12*[+] (ARG1), *asn1*[+] (ASN1), *his4*[+] (HIS7), *pas1*[+] (PCL5), *aes1*[+] (YH19), and *snz1*[+] (SNZ1), involved in amino acid metabolism. This is perhaps more significant considering that only 15 of the Fil/Gcn4-26 genes are in the 3AT-Fil1 set of 573 genes, and that only 5 of the Fil/Gcn4-26 genes are in the 3AT-GAAC set of 68 genes (S7 Table).

The overlap of the Gcn4/Fil1-26 core genes with the *sla1Δ* EMM-Pro set is more highly significant (p < 7.186e[-9]) than their overlap with the *sla1Δ* EMM-NH4[+] set (9.4e[-6]) and with 24.7 vs. 16.7 RFs (Fig 1E). This is consistent with the idea that the *sla1Δ* EMM-Pro transcriptome reflects a basal core constituitive GAAC response to the absence of La whereas the NH4[+] transcriptome reflects an additional stress response.

As part of the response to 3AT, Fil1 appears to activate *pcr1*[+] and *atf1*[+] and other factors that contribute to expression of CESR genes [47]. Prior to discovery of *fil1*[+], the two transcriptomes were segregated using *gcn2Δ* and other mutants, into 3AT-CESR and 3AT-GAAC genes (Fig 1), referred to as Cluster 2 and Cluster 3 respectively in [46]. Although it is beyond the scope of the present study, future work shall assay Fil1, Gaf1 and Atf1 expression in *sla1Δ* vs. *sla1*[+] in various media.

## A case for nuclear surveillance signaling as one component of the GAAC response in *sla1Δ* cells

The GAAC response is activated by uORF-dependent translation of GCN4 mRNA in *S. cerevisiae* either by the classic Gcn2-dependent pathway or by alternate paths independent of Gcn2 and phosphorylation of Ser-51 of eIF2α. Nuclear surveillance is triggered by accumulation of nuclear unprocessed aberrant pre-tRNA and activates a Gcn2-independent pathway of the GAAC response in *S. cerevisiae* [17, 18, 21–23] and can be rescued, reversed or prevented by expression of nuclear tRNA maturation factors including La protein and the tRNA nuclear exporter, Los1 [17, 22].

Nuclear pre-tRNA processing is constitutively perturbed in *sla1Δ* cells. It may be argued that this could lead to overall lower levels of some tRNAs. A number of mature tRNAs accumulate at lower levels in *sla1Δ* relative to WT cells in EMM-NH4[+], some as low as 65% [5]. Hypoacylation was documented for tRNA[Leu], tRNA[Lys] and tRNA[Ser], at 0.80–0.86 levels in *sla1Δ* cells relative to WT, whereas all others were at ≥0.97 in EMM-NH4[+] media [5], but tRNA acylation was at ≥0.97 in YES media (does not contain NH4[+]). Notably, this limited hypoacylation appeared highly selective, in each case for only one tRNA isoacceptor in its

family reflecting its differential sensitivity to misfolding in the absence of La [5]. As noted *sla1Δ* cells exhibit restrictive uptake of leucine and other amino acids regulated by Any1 (below) that is exacerbated by NH4+ [50], and this may help explain why the tRNAs were hypoacylated in EMM-NH4+ [5].

Deletion of *LOS1* from *S. cerevisiae* which causes nuclear pre-tRNA accumulation and interferes with the maturation of numerous tRNA species, leads to a GAAC response with no detectable inhibition of general translation, in an apparent Gcn2-independent manner [35]. It is remarkable that ectopic expression of *los1+* in *sla1Δ* cells completely reversed activation of the three GAAC genes examined, also rescued slow growth, and restored a more normal balance of nuclear end-containing and pre-export pre-tRNAs [50] and here, Fig 2 demonstrates that *rrp6* deletion decreases GAAC gene expression in response to *sla1Δ*. Because a primary defect in *sla1Δ* cells is general perturbation of nuclear pre-tRNA processing [20, 50, 84, 85, 88], this would appear to reflect a connection between tRNA nuclear surveillance and a GAAC response akin to *S. cerevisiae* [22]. As noted in the Results section regarding Fig 2, deletion of *RRP6* from *S. cerevisiae* and *rrp6+* from *S. pombe* have different effects on growth, the latter in a NH4+ media-specific manner. Also, deletion of *RRP6* rescued a nuclear surveillance phenotype that could also be rescued by ectopic expression of tRNAi^Met presumably reflecting that the growth deficiency was due to degradation of a single pre-tRNA species related to the known unique sensitivity of pre-tRNAi^Met to lack of m$^1$A58 in the *trm6-504* mutant [23]. The growth deficiency of *sla1Δ* cells is more complex. In any case, although it is presently beyond the scope of this study, it will be important for future work to determine if deletion of *fil1+* or *gcn2+* will also suppress the GAAC mRNAs (e.g., in Fig 2).

In summary, transcriptome analysis has shown that *sla1Δ* cells are in a state of constitutive basal GAAC activation in all growth media tested. In addition, NH4+ in the media is associated with upregulation of CESR genes in these *sla1Δ* cells. However, even in *sla1Δ* cells growing in YES and EMM-Pro there may be multiple potential triggers of the GAAC, consistent with lines of evidence which indicate that La protein works with tRNA modification enzymes and in some cases synthetases, and its absence can exacerbate deficiencies [4, 5, 20, 92]. Recent results in *S. pombe* suggest the possibility that pertubations to tRNA homeostasis could affect multiple pathways of *gcn2+*-mediated stress signaling [93]. Because La is a general factor involved in nuclear pre-tRNA processing and modification, pertubations in *sla1Δ* cells have potential to disrupt aspects of translation connected to signal pathways beyond *gcn2+*, such as TORC1 [17, 22, 23, 32, 39, 41, 57, 94].

## Spontaneous *sla1Δ* revertants (SSRs) of slow growth relieve inhibition by NH4+ on leucine uptake

The second objective of this study was to isolate *sla1Δ* mutants that revert to a growth rate comparable to *sla1+* on NH4+ media whose characterization advance our understanding of and insight into the biology involved. Although *sla1Δ* cells are deficient for leucine uptake, their slow growth can not be overcome by increasing the concentration of leucine in EMM, consistent with it its uptake inhibition by NH4+ [50]. A striking characteristic was an apparent communal effect of the SSR mutants on the growth of their otherwise isogenic neighbors [95, 96]. As the SSRs emerged as rare mutants from the *sla1Δ* cells they were surrounded by smaller sattelite colonies (Fig 3A and 3B). This is consistent with the hypothesis of radial depletion of NH4+ from the media due to consumption by the SSR accounting for relief of growth inhibition of the surrounding *sla1Δ* cells. Anabolic consumption of NH4+ by a growing SSR would be consistent with the activities of Gdh2 and the multiple other GAAC upregulated genes in *sla1Δ* cells involved in basic amino acid metabolism. This is plausible because as

NH4+ is the only nitrogen source here its benefit for growth would at some point be reduced to a concentration that no longer inhibits amino acid uptake [53, 97]. We note that an alternative hypothesis is that the SSRs release a compound/molecules that neutralize or otherwize override the negative effects of NH4+. Microbial communal life is complex [98, 99], and when affected by NH4+ involves PKA, TOR and SCH9 pathways [100].

## A F32V mutation in SSR5 relieves a negative activity of *any1*+ that appears distinctive to *sla1Δ* cells

A nucleotide mutation that resulted in F32V in the ORF of Any1, the β-arrestin-related endocytic adaptor that was known to negatively regulate leucine uptake in NH4+ media [58, 77] was found in SSR5. The F32V mutation distinguished an activity in *sla1Δ* cells in EMM-NH4+. While *any1-F32V* largely restored growth to *sla1Δ* in NH4+ in SSR5, this allele had no apparent effect on growth in YES in any of the backgrounds tested (Fig 4B and 4C). Also, while *any1*+ was inhibitory to *sla1Δ* growth in NH4+, *any1-F32V* relieved this inhibition, whereas both alleles supported the *any1*+ function for growth in YES (Fig 4B and 4C).

Thorough studies have shown that Any1 and Pub1 work together to negatively regulate trafficking of amino acid transporters Aat1 and Cat1 to the plasma membrane [77]. The *S. pombe* Tsc1 and Tsc2 homologs of the tuberous sclerosis proteins regulate the amino acid transporter system upstream of Any1 and Pub1. Nakase et al. have shown that Any1 and Aat1 physically interact *in vivo* and can be co-immunoprecipitated as epitope-tagged proteins [77]. Finally, conditions in Δ*tsc* cells lead to increased stability of Any1-Pub1 interactions resulting in retention of Aat1, its prevention from reaching the plasma membrane. We previously showed that deletion of *pub1*+ suppressed slow growth of *sla1Δ* in EMM in NH4+ and deletion of *tsc1*+ exacerbated it [50]. According to the model pathway proposed by Nakase and coworkers we suppose that the *Any1-F32V* mutation weakens retention of Aat1 by the Any1-Pub1 system under conditions that are sensed as nitrogen starvation in *sla1Δ* in EMM in NH4+ media. The *Any1-F32V* allele may be useful in future studies that seek to understand mechanisms involved in such regulation.

## Materials and methods

### Yeast strains and growth media

Yeast strains are listed in Table 1. The *leu1*+ derivative of yAS113 was made by integrating pJK148 [101] at the *leu1-32* locus. CY1697 and CY1698 were made by replacing *any1*+ with *any1::kan MX6* in yAS113 and yAS99 respectively. CY1710 and CY1712 were made by integrating *any1*+ *pcr2.1NAT* into CY1697 and CY1698 respectively. CY1711 and CY1713 were made by integrating *any1-F32V pcr2.1NAT* into CY1697 and CY1698 respectively.

Media were prepared according to standard recipes. The recipe for standard EMM is described in [50]; according to EMM "complete" containing additional leucine, adenine, and uracil, each at 225 mg/l, unless selection for a plasmid was required. For EMM-Pro, the NH4Cl was replaced with 10 mM proline.

### cDNA microarray analysis

Yeast growth, RNA isolation, cDNA labeling, microarray hybridization, data processing and the quantified data obtained for the three growth media after normalization were reported [50]. Dr. Juan Mata (Cambridge) kindly provided the 26 core genes bound by Fil1 and Gcn4 [47] that comprise the Fil1\Gcn4-26 gene set. The 3AT-GAAC and 3AT-CESR gene sets are from [46]. Other gene data are available on request. Statistical significance of overalapping genes was by http://nemates.org/MA/progs/overlap_stats_prog.html.

**Table 1.**

| *S. pombe* Yeast Strain | Genotype | Source or reference |
|---|---|---|
| yAS99 (WT) | h⁻ ade6-704 leu1-32 ura4D | [85] |
| yAS113 (*sla1Δ*) | h⁻ ade6-704 leu1-32 sla1Δ::ura4⁻FOA**ᴿ** | [85] |
| CY1671 (*sla1Δ leu1⁺*) | h- ade6-704 leu1-32::*leu1⁺* ura4D sla1⁻ | This study |
| CY1664 (SSR4) | h⁻ ade6-704 leu1-32 ura4D sla1Δ (unknown suppressing mutation) | This study |
| CY1665 (SSR5) | h⁻ ade6-704 leu1-32 ura4D sla1⁻ any1-F32V | This study |
| CY1666 (SSR6) | h⁻ ade6-704 leu1-32 ura4D sla1⁻ del chr II:3487055–3487059 | This study |
| yYH7a | h⁻ ade6-704 ura4D leu1-32 rrp6Δ::kanMX6 | [20] |
| yYH8a | h⁻ ade6-704 ura4D leu1-32 sla1Δ::ura4-FOA**ᴿ** rrp6Δ::kanMX6 | [20] |
| CY1697 | h⁻ ade6-704 leu1-32 ura4D sla1⁻ any1::kan MX6 | This study |
| CY1698 | h⁻ ade6-704 leu1-32 ura4D sla1⁺ any1::kan MX6 | This study |
| CY1710 | h⁻ ade6-704 leu1-32 ura4D sla1⁻ any1::kan MX6 any1⁺ | This study |
| CY1711 | h⁻ ade6-704 leu1-32 ura4D sla1⁻ any1::kan MX6 any1-F32V | This study |
| CY1712 | h⁻ ade6-704 leu1-32 ura4D sla1⁺ any1::kan MX6 any1⁺ | This study |
| CY1713 | h⁻ ade6-704 leu1-32 ura4D sla1⁺ any1::kan MX6 any1-F32V | This study |

## Gene ontology (GO)

Lewis-Sigler Institute, Princeton, https://go.princeton.edu/cgi-bin/GOTermFinder [59], PomBase GO terms.

## Plasmids

The *any1⁺* and *any1-F32V* integrative plasmids were constructed by PCR from yAS113 and SSR5 respectively with primers Any1F: SpeI GATCATACTAGTCGTAACTTCTATGCGCCTCA and Any1R: PmeI ACCTATAGTTTAAACTGCCGAAGGCATATTACTCA generating the *any1⁺* ORF plus ~900 bp upstream and ~220 bp downstream genomic sequence. These were digested with SpeI and PmeI and integrated into the *pcr2.1NAT* vector. Resulting plasmids were digested by NdeI for integration.

## RNA purification and northern blotting

Total RNA was purified as described [102]. For blotting, 5 and 10 μg of total RNA were separated in 1% denaturing agarose gel, transferred to nylon membrane (GeneScreen Plus; Perkin-Elmer), UV cross-linked, baked and hybridized at 42˚C overnight with random primed **³²**P-DNA fragments of genes of interest. Hybridization solution was 5X Denhardt's, 5X SSC, 50% formamide, 0.2% SDS, 5 mM EDTA, 100 μg/ml total yeast RNA (Baker's yeast, Sigma). Quantitation was by PhosphorImager FLA-3000 (Fuji Film). For sequential probings, signal was stripped, monitored for loss of **³²**P before next probing.

## Whole genome sequencing (WGS)

DNA isolated from *S. pombe* was prepared as a multiplexed library and subjected to paired-end sequencing generating 100-bp reads on an Illumina HiSeq. Whole genome read coverage was ~39X for SSR5 and higher for SSRs 4 and 6. The chr II:3487055–3487059 (minus strand) 4 bp deletion at the *zfs1⁺* (SPBC1718.07c) locus converts 5'-AAATAAAATAC-3' to 5'-AAATAC-3'.

**Alignment of WGS library reads.** Read pairs were aligned without relative position constraints using Bowtie2 software [103] to the reference genome, yAS113, *sla1Δ* [76] with the

following modifications as described: the *sla1*⁺ locus had been deleted and replaced with the disrupted *ura4* locus derived from sequencing of the parent construct, base mutations previously noted by the Broad Institute were incorporated to bring genomic sequence into agreement with the FS101 background, and the chromosome 2 N-gap was filled with resolved sequence [76].

**WGS mutation calling.** Single-base as well as insertion or deletion mutations between reference sequence and SSR strains were called using *SAMtools* pileup [104] noting sites where >40% base identity altered and attaching any relevant annotation or coding domain changes with ANNOVAR [105] as previously described [76]. Copy number variation was predicted from averaging mapped read depth across a sliding 400-base pair window while normalizing datasets to average overall read depth. Copy number variations were flagged at locations where this change exceeded 0.8x that of average depth.

**Verification of WGS mutations.** Mutations were verified from DNA prepared from acid-washed bead-lysed *S. pombe* cells. Lysate was treated with DNase-free RNase A. Following phenol/chloroform extraction, pH 7.9 DNA was ethanol precipitated and quantitated. PCR amplification of regions of interest was performed, purified using QIAGEN QIAquick PCR cleanup kit and products sequenced by Macrogen USA Inc to verify mutations.

## ³H-leucine uptake

This was performed as described, in EMM-NH4⁺ media containing 225 mg/l leucine (~1.6 mM) and a trace amount of ³H-leucine [50, 51]. Multiple regression/correlation analysis to derive p-values and t-tests from the ³H-leucine uptake data were as described [106].

## Supporting information

**S1 Table. Different media-dependent *sla1Δ* up-regulated gene transcriptome categories.** (XLSX)

**S2 Table.** A. GO analysis: *sla1Δ*-com = 1.5x up in both EMM NH4⁺ and EMM Pro (38 genes). B. GO analysis: *sla1Δ*-all = 1.5x up in YES, Pro and NH4⁺ (23 genes). (ZIP)

**S3 Table.** A-C: GO analyses: *sla1Δ*-λYES (11 genes), *sla1Δ*-λPro (22 genes), *sla1Δ*-λNH4⁺ (26 genes). (XLSX)

**S4 Table. GO analysis of gene transcripts >1.5x up, *sla1Δ* in EMM-Pro (60 genes).** (XLSX)

**S5 Table. GO analysis of gene transcripts >1.5x up, *sla1Δ* in EMM-NH4⁺ (64 genes).** (XLSX)

**S6 Table. GO analysis of gene transcripts >1.5x up, *sla1Δ* in YES (40 genes).** (XLSX)

**S7 Table. 26 orthologs in *S. pombe* & *S. cerevisiae* bound by Fil1 & GCN4.** (XLSX)

**S1 Raw image. A PDF file containing raw images of the northern blots for Figs 2, 3 and 5 as required by the journal.** (PDF)

## Acknowledgments

The authors would like to thank Alan Hinnebusch for discussion and very helpful critical comments on the manuscript, Juan Mata for sharing the ID names of the 26 orthologous genes bound by GCN4 and Fil1, Steve Coon and colleagues at the NICHD Molecular Genomics Core facility for sequencing.

## Author Contributions

**Conceptualization:** Vera Cherkasova, Richard J. Maraia.

**Data curation:** James R. Iben, Kevin J. Pridham, Richard J. Maraia.

**Formal analysis:** Vera Cherkasova, James R. Iben, Kevin J. Pridham, Alan C. Kessler, Richard J. Maraia.

**Investigation:** Vera Cherkasova, Kevin J. Pridham, Richard J. Maraia.

**Methodology:** Vera Cherkasova.

**Project administration:** Richard J. Maraia.

**Resources:** Richard J. Maraia.

**Supervision:** Richard J. Maraia.

**Writing – original draft:** Vera Cherkasova, Alan C. Kessler, Richard J. Maraia.

**Writing – review & editing:** Vera Cherkasova, James R. Iben, Alan C. Kessler, Richard J. Maraia.

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
