## [Decision Letter · Decision Letter 0]

1 Apr 2021

PONE-D-21-06328

The leucine-NH4+ uptake regulator Any1 limits growth as part of a general amino acid stress response to loss of La protein by fission yeast

PLOS ONE

Dear Dr. Maraia,

Thank you for submitting your manuscript to PLOS ONE. After careful consideration, we feel that it has merit but does not fully meet PLOS ONE’s publication criteria as it currently stands. Therefore, we invite you to submit a revised version of the manuscript that addresses the points raised during the review process.

I have received reports from two expert referees, and both recommend that the paper is acceptable for publication after minor revision, including some editorial and statistical reevaluation of the data and the manuscript.

We look forward to receiving your revised manuscript.

Kind regards,

Reiko Sugiura, M.D., PhD.

Academic Editor

PLOS ONE

Journal Requirements:

[This work was supported by the Intramural Research Program (HD000412-31PGD) of the Eunice Kennedy Shriver National Institute of Child Health and Human Development, National Institutes of Health. The funders had no role in study design, data collection, interpretation, or decision to submit for publication.]

 [The funders had no role in study design, data collection and analysis, decision to publish, or preparation of the manuscript.]

5. Please amend either the title on the online submission form (via Edit Submission) or the title in the manuscript so that they are identical.

6. Please include your tables as part of your main manuscript and remove the individual files. Please note that supplementary tables (should remain/ be uploaded) as separate "supporting information" files.

Reviewers' comments:

Reviewer's Responses to Questions

**Comments to the Author**

1. Is the manuscript technically sound, and do the data support the conclusions?

Reviewer #1: Yes

Reviewer #2: Yes

2. Has the statistical analysis been performed appropriately and rigorously? 

Reviewer #1: No

Reviewer #2: Yes

3. Have the authors made all data underlying the findings in their manuscript fully available?

Reviewer #1: Yes

Reviewer #2: Yes

4. Is the manuscript presented in an intelligible fashion and written in standard English?

Reviewer #1: Yes

Reviewer #2: Yes

5. Review Comments to the Author

Reviewer #1: In the manuscript submitted by Cherkasova et al, the authors provide the evidence that the sla1+ gene of S. pombe, which encodes La protein, mediates general amino acid control (GAAC) response concomitant with nuclear surveillance mechanisms. First, transcriptome analysis reveals that genes upregulated in sla1D cells exhibit highly significant overlap with GAAC genes. Second, GAAC gene upregulation is suppressed by rrp6 deletion. Third, the authors isolated sla1D spontaneous revertant (SSR), which suppresses slow growth phenotype in NH4 + -media, and identified an F32V mutation of any1+ gene in the SSR mutants by whole genome sequencing. Furthermore, 3H-leucine uptake of SSR- any1-F32V cells in NH4 + -media is more robust than by sla1D cells.

The manuscript presents noble findings regarding a function of sla1+ gene, and is suitable for PLOS ONE. However, several important information is missing as described below, therefore, I cannot support publication of the manuscript in its present form.

Specific comments:

1. The authors showed that sla1D cells exhibited increased levels of gdh2+ and aca1+ mRNAs in EMM-NH4+ -media, which has already been shown by the authors’ previous study (in ref [50]). And then, they showed that additional rrp6D (Fig 2) and any1D (Fig 5) mutations reduced the upregulation. The levels of the increase of gdh2+ mRNA in sla1D cells is not comparable between the experiments in Fig 2 and Fig 5; it was detected at approx. x8 (Fig 2B, right) and x3.6 (Fig 5, lanes 3-4), respectively. It is also the case for aca1+ mRNA; it was detected at approx. x6 (Fig 2B, right) and x4.5 (Fig 5, lanes 3-4). Is there any difference in experimental conditions between these experiments? Is this just within the experimental variations? The authors need to perform these experiments at least three times and present the results with standard deviations and p-values.

2. Expression levels of rpl8+ mRNA in sla1D cells grown in EMM-NH4+ -media is drastically reduced in the experiment in Fig 2A (lanes 11 and 12), but not in Fig 3E (lanes 3 and 4) and Fig 5 (lanes 3 and 4). If the authors perform these experiments under the same conditions, the difference should not be appeared.

3. It seems that the authors isolate five SSRs (SSR1-5), whose growth phenotype was shown in Fig 3C. SSR1 also shows robust suppression phenotype in addition to SSR4, SSR5 and SSR6. Why do they abandon further analysis for SSR1 including whole genome sequencing? The authors should describe the precise clone number of isolated SSR and the reason why they pick SSR4, SSR5 and SSR6 in the manuscript.

4. SSRs were indicated like “SSR-4b” but not “SSR4” in Fig 3C. What is the difference between SSRX and SSR-Xb? The authors should describe the difference in the figure legend or main text.

5. In Fig 5, the authors claimed that any1-F32V led to lower aca1+ mRNA levels than any1+ in the sla1D background (compare lanes 15-16 and 17-18, x3.6 and x2.1, respectively). However, the aca1+ mRNA levels of sla1D any1D double mutant that is integrated sla1+ is also lower than that of sla1D (compare lanes 3-4 and 15-16, x4.5 and 3.6, respectively), which should be comparable, considering the authors claim as described in lines 405-407. The result is not supportive. To confirm the claim, the authors need to present the results with standard deviations and p-values as described above and reevaluate the result.

6. Point mutation sometimes effects on the stability of the protein. The authors need to confirm any1+ and any1-F32V protein levels, in addition to the mRNA expression levels by northern blotting shown in Fig 5.

7. The authors conclude that sla1D mutant exhibited decreased leucine uptake in EMM-NH4+ media compared to WT, in addition, any1D mutant, sla1D any1D double mutant, and SSR5 exhibited greater uptake than WT (lines 412-414). To confirm these differences, again, the authors need to present the results with standard deviations and p-values.

Reviewer #2: The manuscript entitled "The leucine-NH4+ uptake regulator Any1 limits ..." by Cherkasova et al. first analyzed gene expression profiles of a sla1∆ mutant, lacking a tRNA processing factor La homologue in S. pombe, grown in three different N-source conditions and found that the mutant exhibits significant similarity to the wild-type cells under GAAC in transcriptome. In addition, the sla1∆ cells in NH4+ media showed up-regulation of a set of genes that are a part of the CESR-induced genes. The former regulation seems to be driven by nuclear tRNA surveillance since nuclear exosome inactivation by rrp6∆ dampens the up-regulation of the GAAC-related genes in the sla1∆ background. The authors stepped forward to isolate spontaneous suppressor mutants and found that a F32V mutation on the any1+ gene, encoding an arrestin-homologue, suppressed poor growth of the sla1∆ mutant on the NH4+ plate and supported growth of the surrounding any1+ cells. The any1-F32V sla1∆ cells incorporated Leu more efficiently even in the presence of NH4+ than the sal1∆ cells. The authors concluded that the tRNA processing factor La is involved in the GAAC response and Any1 specifically acts to support this signal transduction.

All the experiments seem to be performed technically rigorous and fit to the scientific standard of this field. Essentially, the data presented as figures and tables support the authors' notions. For example, they precisely described discrepancy between rrp6∆ effects on GAAC gene expression and those on growth in NH4+ media, which led to identification of a specific allele of any1+ as a suppressor of sla1∆ with a unique features. They appropriately cited previous paper especially in the transcriptome analyses of the sla1∆ cells. Thus, the reviewer essentially supports the publication of the manuscript in the journal of PLOS One. Before publication, the following minor points should be amended:

1) p. 23, line 581, p. 24, line 584; ug should be µg (micro gram).

2) p. 24, lines 583–584; it says that the Hybridization solution contained 100 µg/ml yeast RNA. Was the RNA prepared from budding yeast but not from fission yeast? Or, it might be "tRNA" but not "RNA."

3) p. 24, line 585; the company name should be "Fuji Film."

4) The gene names and boxes in Tables 1–7 are color-coded. However, there is no explanation of the colors.

5) It is reader-friendly if the standard gene name of every gene listed in the lower section of Tables 2–7 is indicated. In addition, the reviewer is skeptical that the current style of the Tables 1–7 is appropriate for publication even on line. They should be adequately reshaped for publication.

6) In Fig. 2B, it is difficult to recognize bar identity from pattern examples in its inset because the pattern examples were enlarged too much.

7) In Fig. 6, the unit of the axis should be "pmol/10^7 cells."

6. PLOS authors have the option to publish the peer review history of their article (what does this mean?). If published, this will include your full peer review and any attached files.

Reviewer #1: No

Reviewer #2: **Yes: **Tohru Yoshihisa

---

## [Decision Letter · Decision Letter 1]

24 May 2021

PONE-D-21-06328R1

The leucine-NH4+ uptake regulator Any1 limits growth as part of a general amino acid control response to loss of La protein by fission yeast

PLOS ONE

Dear Dr. Maraia,

Thank you for submitting your manuscript to PLOS ONE. After careful consideration, we feel that it has merit but does not fully meet PLOS ONE’s publication criteria as it currently stands. Therefore, we invite you to submit a revised version of the manuscript that addresses the points raised during the review process.

Two experts in the field have now reviewed your manuscript.

As you will see in their comments, both reviewers highly evaluated your revised manuscript. However, Reviewer 1 still raises concerns about your manuscript from a statistical viewpoint, which might require reconsideration of your conclusion of the manuscript.

We look forward to receiving your revised manuscript.

Kind regards,

Reiko Sugiura, M.D., PhD.

Academic Editor

PLOS ONE

Journal Requirements:

Reviewers' comments:

Reviewer's Responses to Questions

**Comments to the Author**

1. If the authors have adequately addressed your comments raised in a previous round of review and you feel that this manuscript is now acceptable for publication, you may indicate that here to bypass the “Comments to the Author” section, enter your conflict of interest statement in the “Confidential to Editor” section, and submit your "Accept" recommendation.

Reviewer #1: (No Response)

Reviewer #2: All comments have been addressed

2. Is the manuscript technically sound, and do the data support the conclusions?

Reviewer #1: Yes

Reviewer #2: Yes

3. Has the statistical analysis been performed appropriately and rigorously? 

Reviewer #1: No

Reviewer #2: Yes

4. Have the authors made all data underlying the findings in their manuscript fully available?

Reviewer #1: Yes

Reviewer #2: Yes

5. Is the manuscript presented in an intelligible fashion and written in standard English?

Reviewer #1: Yes

Reviewer #2: Yes

6. Review Comments to the Author

Reviewer #1: Cherkasova et al. have provided convincing response to issues raised in the previous round of review. I will support its publication after addressing the point as described below.

The authors described in lines 437-440 that “Although these quantifications do not carry strong statistical significance, visual inspection of the northern blot internal controls provide additional evidence that the any1-F32V allele is more effective at suppressing aca1+ levels than is the any1+ allele.” Generally, “visual inspection” sometimes misleads conclusions. I strongly recommend the authors evaluate the result using statistical parameters, such as standard deviations, p values, and t-tests. If the authors cannot find statistical significance of aca1+ mRNA levels between lanes 15-16 and 17-18, they should reconsider the conclusion; “The data suggest that any1-F32V is partially effective as its integration suppresses aca1+ levels to greater extent than any1+.” in lines 444-445.

Reviewer #2: (No Response)

7. PLOS authors have the option to publish the peer review history of their article (what does this mean?). If published, this will include your full peer review and any attached files.

Reviewer #1: No

Reviewer #2: **Yes: **Tohru Yoshihisa

---

## [Author Response · Author response to Decision Letter 1]

31 May 2021

I am sorry that my previous revision did not satisfy one point. On the other hand I am pleased to note that I believe that the several other revisions such as addition of quantification of multiple data points/sets, addition of p-values, R2 values and SDs helped strengthen the paper, and I thank the reviewer and the review process for it.

Reviewer #1 noted “Cherkasova et al. have provided convincing response to issues raised in the previous round of review.” The Reviewer noted that publication would be supported after one point would be addressed, and referred to our description of data on lines 437-440 discussing the quantifications in Fig 5. The short paragraph concluded with “If the authors cannot find statistical significance of aca1+ mRNA levels between lanes 15-16 and 17-18, they should reconsider the conclusion…. in lines 444-445.” 

Reponse: Consideration of this issue by myself and coauthors most knowledgeable on it led to our revision of the manuscript as suggested. The text that had been on lines 437-440 and conclusion on lines 444-445 were deleted. In addition, quantification numerals on lines 436-437 were also deleted. Additional edits to assimilate and accommodate the suggested considerations led to the revised manuscript in which the total extent of edits are limited to lines 431-474.

---

## [Editor Report · Decision Letter 2]

7 Jun 2021

The leucine-NH4+ uptake regulator Any1 limits growth as part of a general amino acid control response to loss of La protein by fission yeast

PONE-D-21-06328R2

Dear Dr. Maraia,

We’re pleased to inform you that your manuscript has been judged scientifically suitable for publication and will be formally accepted for publication once it meets all outstanding technical requirements.

Kind regards,

Reiko Sugiura, M.D., PhD.

Academic Editor

PLOS ONE

---

## [Editor Report · Acceptance letter]

9 Jun 2021

PONE-D-21-06328R2 

The leucine-NH_4_^+^ uptake regulator Any1 limits growth as part of a general amino acid control response to loss of La protein by fission yeast 

Dear Dr. Maraia:

I'm pleased to inform you that your manuscript has been deemed suitable for publication in PLOS ONE. Congratulations! Your manuscript is now with our production department. 

Kind regards, 

on behalf of

Dr. Reiko Sugiura 

Academic Editor

PLOS ONE